# "They organized women who advised me": Adolescents' perceptions of Social and cultural influences on mental health during the postnatal period in Malawi: Qualitative study

Chimwemwe Tembo [1,2]*, Linda Portsmouth[2], Sharyn Burns[2]

**1** Saint John of God Hospitaller Services Malawi, Lilongwe, Malawi, **2** Faculty of Health Sciences, School of Population Health, Curtin University, Perth, Australia

\* chimweptembo@yahoo.co.uk

**Data Availability Statement:** Curtin University survey office, which manages the Human Research

## Abstract

The postnatal period is a critical transition period for first-time mothers, especially adolescents. An indication of maladaptation with this transition is the experience of poor mental health by some adolescent mothers. Social and collective cultural factors have been claimed to influence maternal mental health during the postnatal period; however, there is limited research focusing on cultural and social factors that influence adolescents' mental health during the postnatal period in rural Malawi. Therefore, we explored the perceived cultural and social factors that influence mental health among adolescent mothers during the postnatal period in rural Malawi. A descriptive qualitative design (DQ) was employed. Data were collected from September 7th, 2021, to March 31st, 2022, from a convenience sample of 395 adolescent mothers. Reflexive thematic analysis was employed using open coding. Codes were grouped into categories and themes derived using inductive thinking. Adolescent mothers' mental health was found to be influenced by interactions between social and cultural factors within all levels of the Social Ecological Model of Health (SEM). Four main themes, *health and mental health awareness*, *relationships*, *family support*, and *rite of passage*, explained social and cultural influences for adolescent mothers in Malawi. There is a need for collaborative, coordinated, and co-designed efforts to implement integrated, comprehensive, and culturally appropriate interventions at each level of SEM influence to promote adolescent' mothers' mental health. Specifically, policies should focus on the prevention of early marriages, improved access to reproductive health services, and long-term interventions to promote family involvement in the care of mothers. Healthcare systems should offer accessible mental health support for adolescent mothers.

Ethics Committee for researchers, allows limited access to data generated for this study to protect participants' privacy and related confidentiality agreements. However, interested researchers may reach out to the Human Research Ethics Committee (hre@curtin.edu.au), or the chairperson of the Human Research Ethics Committee and the corresponding author, Mrs. Chimwemwe Tembo (chimweptembo@yahoo.co.uk), to request access to data if needed to reproduce the article or review results.

**Funding:** The authors received no specific funding for this work.

**Competing interests:** The authors have declared that no competing interests exist.

## Background

The World Health Organization (WHO) has estimated that 11% of births globally are to adolescent girls aged 15–19 years [1]. These rates are higher in sub–Saharan Africa (SSA), where about 20–40% of adolescents are either mothers or are pregnant. In Malawi, 31% of all births are from adolescent mothers [2,3]. It is estimated that 46%of girls are married before the age of 18, and 9 percent before the age of 15 [4–6], contributing to higher rates of adolescent childbearing [7]. To address these rates, the Marriage, Divorce, and Family Relations Act, Chapter 25:01 of the Malawi Constitution, was amended in 2017 and raised the age of marriage to 18 years for all genders [8]. The government of Malawi is collaborating with other non-governmental organizations to implement strategies to end child marriages in an effort to delay childbearing [9]. However, while the amendment has had an impact within urban areas, adolescent girls in rural areas continue to be exposed to traditional cultural practices that perpetuate child marriage and early childbearing. An example of such cultural practices includes exposure to sexual activities at initiation ceremonies [3,6,10]. These initiation ceremonies also provide young girls with misinformation about sex and marriage [11] encouraging early initiation of sexual activity in an environment where there is poor uptake and availability of contraceptives [12].

Adolescent motherhood may result in social and psychological consequences associated with coping with motherhood while transitioning from childhood to adulthood [13,14]. Significant evidence indicates that adolescent childbearing presents a range of physical and mental health risks for the adolescent and her infant and may result in serious societal, developmental, and economic problems [15–17]. For example, in Malawi, adolescent motherhood is associated with higher infant mortality, stunted growth, low school enrolment and poverty [3]. In South Africa, when compared with the adult population, adolescent mothers present with an increased risk for common maternal mental health problems such as anxiety and depression [17]. Globally, the postnatal period represents a risk with adolescent mothers experiencing significant mental health concerns [18]. In low- and middle-income countries (LMICs), approximately 16% to 40% of perinatal women experience common mental disorders such as anxiety and depression with substantial variations across settings [19]. Some variations are hypothesized to be due to differences in screening tools, such as diagnostic versus self-reporting tools and different cut-off points. For example, the reported prevalence of postnatal depression among adults in a peri-urban in South Africa was 38.8% with an Edinburgh Postnatal depression scale (EPDS) cut-off point of 13 [19,20]. In comparison, a study of perinatal adult mothers in Malawi reported a prevalence of 30.4% using the Self-Reporting Questionnaire SRQ and 21% using the Structured Clinical Interview for DSM-IV (SCID) [21]. Evidence shows that adolescent mothers are more likely to experience common mental disorders compared with adult mothers [22,23]. The rate of postpartum depression in adolescent mothers is estimated to be between 14 and 53% worldwide [24,25]. Furthermore, depressive symptoms persist in adolescent mothers for longer durations after delivery compared with adult mothers [18].

The risk factors for postnatal depression in LMICs include poverty due to socio-economic hardship, lack of emotional and practical support, intimate partner abuse, HIV status, inadequate social and familial support, unintended pregnancy, and a negative societal perception of adolescent pregnancy [22,23,26].

Despite risks and prevalence, a South African study revealed that the majority of adolescent mothers do not seek mental health help due to stigma and are usually not screened when they reach out for maternal health services [27]. Previous international studies have recommended early identification through screening and treatment to prevent the negative consequences of poor mental health [28,29].

Cultural and social factors are believed to impact a mother's mental health. For example, a Nigerian study found that some postnatal mental health disorders can only be fully understood when cultural and social factors are explored [30]. Furthermore, the Nigerian study reported that cultural rituals and other social influences determine the pathway to care, the choice of intervention or support, and patterns of family life in the transfer of knowledge and beliefs between generations. Therefore, the cultural context should inform interventions for postnatal mothers [30].

In Malawi, research on the impact of cultural practices on perinatal mental health among adolescent mothers remains unexplored. In rural areas, adolescent mothers continue to experience cultural practices and rituals that may influence their mental health. This study forms part of a broader study and aims to explore the perceived social and cultural factors that influence the mental health of adolescent mothers in rural Malawi.

## Methods

### Theoretical framing

This study used Bronfenbrenner's Social Ecological Model (1979) as a holistic approach to understanding the experiences of adolescent mothers during the postnatal period [31–33]. The Social Ecological Model was chosen as a framework for understanding the mental health of adolescent mothers because the model suggests that an individual's well-being is influenced by interconnected factors at different levels, including the individual, interpersonal, community, and broader society [31,32]. Furthermore, the Social Ecological Model highlights the significance of addressing social justice and systemic factors that impact an individual's development and well-being [31]. In this study, the model informed the development of interview guides and was applied to understand the results after inductive thematic analysis.

### Study design

A descriptive qualitative design (DQ) was employed to answer the question: what cultural and social factors influence mental health among adolescent mothers in rural Malawi? This research design was chosen to facilitate an understanding of participants' opinions, suggestions, beliefs, and practices [34]. The strength of a qualitative descriptive design is underpinned by the focus on principles of openness, pre-understanding, and adaptation of a reflective attitude [35]. However, instead of using bracketing, we intended to build on questioning as a representative way of describing meaning and understanding from the perspective of the participants [34,36]. The researcher's familiarity with being a Malawian adolescent mother contributed to a more robust understanding of the observed behaviors and interpretation of the interviews. However, it was considered important to omit sharing the researcher's own motherhood background with the participants to avoid any feelings of transference within the relationship between the researcher and the participants.

### Setting

The research was conducted at Mitundu Rural Hospital (MRH) and its outreach clinics. The catchment of MRH is characterized by high levels of poverty, high rates of early marriage, and malnutrition. Mitundu has a population of 147,822 people, with 29,171 households and 402 villages. The catchment area of Mitundu is dominated by the Chewa tribe, which comprises 34% of the Malawian population [37]. The district adolescent fertility rate of women between the ages of 14 and 19 is 165 per 1000 women, with an average of 135 adolescent deliveries per month. The facility has 13 outreach clinics, which are visited once a month.

## Ethics approvals and consent to participate

Ethical approval was received from the Curtin University Human Research Ethics Committee (HRE2021-0223) and the Malawian Ethics Board National Committee on Research Ethics in the Social Sciences and Humanities (P.05/21/575). Adolescent mothers were regarded as emancipated minors; approval was granted to apply mature minor status.

## Participant recruitment and sampling

The study participants were adolescent mothers aged 19 and younger who were less than 12 months in the postnatal period and had the capacity to consent. Participants were conveniently recruited at clinics. Recruitment occurred when mothers attended a clinic for postnatal check-ups, under-five clinics, and family planning clinics. Interested adolescents meeting the eligibility criteria were provided with an information sheet written in Chichewa and read to participants by the research assistant before the interview. Additional questions were asked to ascertain whether the participants understood their potential involvement. Both verbal and written consent were provided. Participants who were illiterate and unable to sign their names were requested to use their thumbprint as a signature (this is standard practice in Malawi), and participants were assured that they could withdraw from the study at any time. Participants completed a survey and in-depth interview detailed in previous publications [38,39]. Any woman who reported suicidal ideation was counselled and offered a referral for clinical mental health assessment and treatment. Participants were also given mental health information brochures written in Chichewa.

## Data collection

Participants were initially asked about their general understanding of health and mental health before being asked about specific social and cultural factors. Table 1 provides a summary of questions from the interview guide informed by the Social Ecological Model [40] and a review of published literature. The guide was initially developed in English before being translated and administered in Chichewa. The translation followed the WHO standardised translation process (WHO, 2014). The team of professional independent translators whose first language is Chichewa translated the tools into Chichewa. Three experts in mental health and the principal investigator, who is fluent in both English and Chichewa, worked with professional translators to assess the translated guides to identify and resolve discrepancies and inadequate expressions or concepts, and discrepancies. This was especially important for the translation of terms such as 'mental health,' 'anxiety' and 'depression' into Chichewa. The first author and two research assistants facilitated the one-on-one face-to-face interviews. Where consent for recording was received, interviews were recorded and transcribed verbatim. Approximately 20% of participants did not agree to being audio-recorded. For these participants detailed

**Table 1. Interview guide questions.**

| Interview guide questions |
| --- |
| 1. What do you understand by the terms health and mental health? |
| 2. Tell me more about your experiences during this pregnancy/delivery and the postpartum period. Probes: How do these experiences affect your feelings, thinking and behaviour? |
| 3. What are some cultural and social issues that surround childbirth that influence your mental health during the postnatal period? Probe: What cultural practices do you think are supportive |
| 4. What practices may negatively influence the mental health of adolescent mothers? |
| 5. What is your opinion about mental health support that can best help adolescent mothers in this area? |

notes were taken. Fieldnotes were made for all participants which included observations of the participants expressions and behaviors. The researcher conducted reflections by keeping a reflexive diary during the entirety of the data collection process, during which thoughts, emotions, and observations were recorded while engaging with individuals and gathering data. This served as a resource for analyzing the positionality and potential biases [41,42]. The interviews ranged from 20 to 60 minutes in duration.

### Data analysis

At the end of each day, the research assistants (PK and CN) and the lead researcher (CT) reflected on emerging data and themes through discussions. MG, a secondary school teacher familiar with the data transcription process, transcribed the interviews verbatim. A reflexive thematic analysis method was used to analyze the transcripts and notes, where patterns within the collected data were identified [43,44]. Braun and Clarke's [44] six-phase guide to thematic analysis was employed as a framework. During the first step, familiarization with the data was achieved by reading and re-reading the transcripts and interview notes. Step two involved generating initial codes using open coding, whereby the codes were developed and modified by the researcher (CT) and MZ. The third step involved generating themes according to the patterns identified. Codes were grouped and allocated to a theme. During this process, CT and MZ were coded separately and discussed, and SB then confirmed the analysis. Step four involved reviewing the themes and was conducted by authors CT, SB and LP. The fifth step involved the final review and defining of themes and subthemes [43–46]. The final step was the write-up of the results. Confirmability and dependability were achieved via peer checking by experts in qualitative methods [47]. The supervisory team SB and LP also contributed to the data interpretation and ongoing reviews of the results [47]. Qualitative data were managed and analysed using NVivo version 11 [39,48]. Participants' demographic characteristics were descriptively analyzed using SPSS 27.

## Results

### Participants' characteristics

Demographic characteristics are presented in Table 2. The participants' mean age was 17(SD 1.157) (age range 14 to 19 years). Most participants were married at the time of the interview (n = 308; 78%); 17% (n = 65) were single; and 5% (n = 21) were divorced.

### Overview of the themes

Four themes categorized and described the social and cultural influences on mental health for adolescent mothers in rural Malawi. The overarching theme *health and mental health awareness* helps understand participants' contextual understanding of health and mental health. This understanding also impacts the other three main themes: *relationships*, *family support*, *and rites of passage*. The themes reflected demonstrate the interconnectedness of multiple factors that influence adolescent mothers' mental health during the postnatal period within their social and cultural context. Fig 1, illustrates the association between the social and cultural factors.

### Health and mental health awareness

The overarching theme of health and mental health awareness provides context for the impact of social and cultural factors. Participants' understanding of health reflects learnings and beliefs likely to have been influenced by family and cultural beliefs. Prior to discussing mental

**Table 2. Participants demographic characteristics (n = 395).**

| Participants characteristics | n | % |
|---|---|---|
| **Age in years** | | |
| Under 16 | 64 | 16% |
| 16–17 | 71 | 18% |
| 18–19 | 260 | 66% |
| **Marital status** | | |
| Single | 65 | 16.5% |
| Widow | 1 | 0.3% |
| Married | 308 | 78.0% |
| Divorced | 21 | 5.3% |
| **Married before pregnancy** | | |
| Yes | 113 | 34% |
| No | 217 | 66% |
| **Polygamous marriage** | | |
| Yes | 38 | 9.6% |
| No | 270 | 68.3% |
| **Who the adolescent mother lived with** | | |
| Parents | 85 | 21.5% |
| Husband | 300 | 76% |
| Others | 10 | 2.5% |
| **Schooling** | | |
| No schooling | 5 | 1.3% |
| Primary | 350 | 88.6% |
| Secondary | 40 | 10.1% |
| **Religion** | | |
| Muslim | 4 | 1.0% |
| Christian | 378 | 95.7% |
| Traditional Religion | 13 | 3.3% |

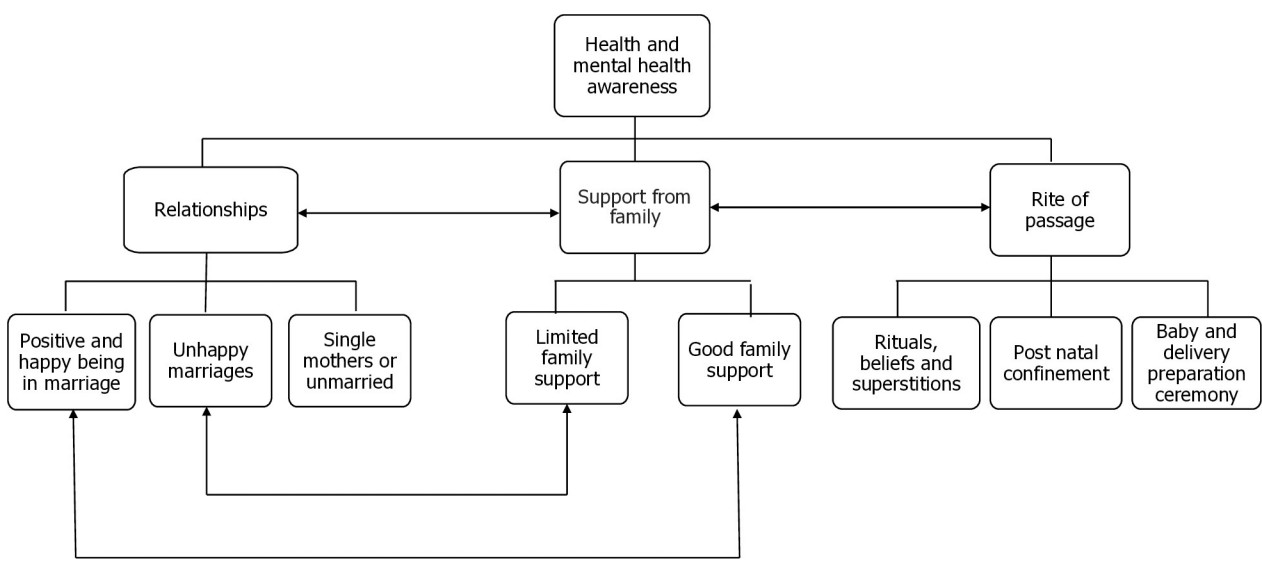

**Fig 1. Social and cultural factors influencing the mental health of adolescent mothers in Malawi during the postnatal period.**

health, participants were asked about their understanding of health in general. Most participants demonstrated limited understanding of general health and did not discuss physical and mental health concurrently. Participants did, however, suggest 'good health' as being physically healthy and being free from sickness, having a good diet, having 'enough blood' and not experiencing any pain. For example, *"Hmm. . .. Health means the person is eating different groups of food. . .An unhealthy person is when the person is thin, and the body is not that good. . . Laughing. . . Hmmm when someone is not sick, the body is strong and eats well. And has enough blood. When you do not feel any pain and you are not sick frequently. Eat well, and you are fat. You are healthy"*. (*17-year-old, married mother*)

However, when participants were specifically asked about mental health, many adolescent mothers described good mental health as a state of having no worries. Most did not explicitly describe mental health; however, they described their personal emotional experiences that affected their well-being during the postnatal period as examples of poor mental health. Some adolescent mothers commented on experiences of excessive worry, hopelessness, lack of interest in anything, guilt and how these impacted their coping. These experiences of emotional distress were described as linked to being unhappy and not necessarily having a serious psychological problem that required treatment., Some adolescent mothers described mental health in relation to their feelings, for example *"I think about my life, most of the times I am happy maybe for two days but then the rest of the times I am thinking about many things, but they are not fulfilled." (18- year- old married mother)*. Another adolescent mother stated "*Am always feeling sad. I am not interested in anything. Even when my friends tell me something good, I still don't feel good. . .. How I will look after the child. I do regret, I. . . eh mm, because having a child is hard, I feel guilt . . .I don't feel good, most of the time I don't have strength to take on daily chores.***" (***14- year- old unmarried mother***).** Some adolescent mothers also described mental health struggles through physical complaints such as experiences of headache, not sleeping, heart palpitations, being isolated, and physical discomforts. For example, one adolescent mother said, *"My heart is burning and I am having a headache.". . .." My heart gets extremely hot. Then I pour water on myself. For instance, the time I came here my heart was very painful" (18-year-old married mother)*. Some participants discussed how having a baby impacted their social and interpersonal lives. The context of these experiences is suggestive of emotional suffering as stated by one adolescent mother. *"It's hard now because I'm trying to juggle school and, a baby;" "I miss my old life, I used to be able to hang out with my friends whenever I wanted, now I can't;" and "I wasn't prepared for how hard it would be, I feel like I' m failing."* (*14-year-old single mother*).

Although participants described a range of mental health issues, many discussed keeping their feelings of psychological distress a secret to avoid being judged, for example: "*I am afraid of nurses who treat us harshly because we did not use family planning. So, I couldn't share what I was thinking and feeling.. . ., so, we usually fail to ask questions even if we have them.* (*19-year-old unmarried mother*).

## Relationships

Many participants reported that issues related to relationships in the context of marriage had impacted their psychological well-being during the perinatal period, with some discussing positive aspects of relationships while others described stressors. Most participants discussed the importance of having a partner, as a provider of personal and financial support. A few adolescent mothers who were single and unmarried wished they were married, especially if the father of the child did not wish to marry them. However, other single mothers lived with their parents and discussed being happy with this arrangement. While some participants described

how marriage positively impacted their mental health, others found marriage stressful and that it generated feelings of uncertainty. Some mothers reported that cultural norms, societal influences, and family expectations shaped adolescent mothers' attitudes towards marriage, and the impact marriage had on their mental well-being, in particular the pressure to have a husband. For many adolescent mothers, their decision to marry was driven by an unplanned pregnancy. Only 34% of participants reported they were married prior to being pregnant with 78% married at the time of the survey. More than half of the participants who were not married at the time of conception described receiving pressure from their families to get married.

The theme of *relationships* has four sub-themes: *positive and happy; being in marriages, unhappy marriages, single mothers or unmarried.*

## Positive and happy being in marriage

Within the sub-theme of happy marriages, participants fell into different categories: planned marriages (those who were already married and were happy to be pregnant); unplanned but wanted marriages (the pregnancy was unplanned, but the marriage was wanted and happy); eloped and married once pregnant and happy; and being unmarried. Many who were already married at the time of pregnancy reported they made the decision about when and whom they wanted to marry, were satisfied with their marriages, and were well prepared to have a baby. These mothers were happy with their married life as they viewed marriage as a means of financial security and a source of happiness, and they were excited to raise a child with their husband. These adolescents generally discussed having good emotional and financial support from their partners. Some adolescents who had an unplanned pregnancy and experienced conflict and initial rejection by their parents or guardians reported that being accepted in a marriage with the father of their child helped them have a sense of independence and reduced the burden of caring for the child. These participants discussed having the support of their partner and felt they had a sense of security and peace to look after the child. For example, *"The way they (my parents) treated me at home was bad. Sometimes they (my parents) did not give me nsima (food) to eat. Most of the time I was very sad. However: The time they took me to stay with the man responsible for the pregnancy some of the problems ended. We stayed peacefully and am happy" (17-year-old married mother)*. Some adolescents found the arrangement of forced marriage to be a relief, and they were happy to be married.

Some adolescent mothers reported that their parents expressed concerns around their marriage related to their age, not approving of the baby's father and wanting them to go back to school after the child's birth. In such cases, some participants discussed going against their parents' wishes, deciding to elope so they could raise the child together and were consequently happy and satisfied with their marriage. *"We agreed to get married, but my parents said that I am still young since I was 14 years old. they wanted me to go back to school after delivery. So, he came during the night and took me, and I ran away from home. I packed my clothing, and we just eloped" (15 year-old -married mother)*. Some participants, who were not interested in school, indicated that they deliberately planned their pregnancy pushing their families to arrange marriage quickly. For example, *"I just decided to become pregnant, because all my friends were married so I also wanted to be married but my parents were refusing". (16-year-old married mother)*.

Happy marriages provided emotional reassurance to mothers and lessened the stress associated with motherhood because it was perceived as an assurance for sharing parenting responsibilities.

## Unhappy marriages

Some adolescent mothers reported issues surrounding societal expectations, rejection, coercion into marriage, and violence as having a negative influence on their psychological well-

being. Participants described how social and cultural norms about traditional marriage arrangements influenced their married life and mental wellbeing especially when the pregnancy was unplanned, and the marriage was forced. Personal conflict between their own self-assertiveness and a lack of power to make an independent decision regarding marriage was discussed. For example, adolescents explained that once it was known to their families that they were pregnant, relatives from the adolescent mother's side reported to the chief's court. The chief ordered the father's relatives to organize a ceremony called "Kaloze", which means "go and point the man". During this ceremony, the adolescent is asked to identify the man responsible for her pregnancy among the group of male community members. Some adolescent mothers reported that this ceremony was embarrassing and pushed them into marriages, while others viewed this ceremony as a social norm and described being married in this way as their own choice and decision. A few reported that to avoid the kaloze ceremony, parents from either the husband's or their own family exercised some urgency to get them married before the community knew that they were pregnant. For example, *"The chief organizes a group which is composed of adolescent girls and women. . . Everyone who wants to come can do so. So, they also invite all boys and men in the village, and they tell you to point at the man who is the one responsible for the pregnancy. All the people surround you on an open ground. . . I was forced to attend this type of ceremony which I did not want. . . and after this ceremony, they just took me to my husband's village where I was living with my mother-in-law, I had no choice and did not want to get married, but I could not say no. I am in an abusive marriage and am not happy"*. (18-year-old married mother). In this community, chiefs are very powerful, and the community listens to them. While no punishment is given specifically to adolescents who refuse to participate in this ceremony, the mother's family must pay a 'leywrite' or a fine to the chief if they don't participate.

Many participants reported that cultural traditions and social pressure influenced their acceptance of the marriage, even when they did not want to get married, resulting in dissatisfaction with their marriage. Some participants described unwanted polygamous marriages and discussed how they had experienced limited support from their husbands to support their babies. For example, two adolescents said *"This man is already married, and he told me that he is married when I told him that I am pregnant. . .(sobbing). I avoided being embarrassed. . .. Yes, it was a polygamous marriage hidden from the public. My parents were not happy with me. They just took me to my husband's village and asked them to marry me". (17-year-old married mother).* Another adolescent who was also forced to get married in a polygamous relationship said *"Look, now they don't send anything, no assistance. So, I am just living here. . . so mostly am not happy. I worry a lot. am worried about the care. More especially, the baby. My husband is always away and does not take part in the care. He lives with his first wife." (An 18-year-old married mother)* and another mother also reported, *"He stopped giving me child support and many other things. . .. I didn't know he was married.)" (18-year-old mother in a polygamous relationship)*

Some participants reported they were in marriages where they experienced violence and were asked to persevere. *"Most of the time, my husband insults me for no reason. And the time I was at my mother's house, they were also insulting me for no reason. . . . He swears at me even when I am with friends; he would call me a dog.. . . I realized this after we got married. People from my husband's side told me about the problem and advised me to ignore him when he behaves strangely and not take him seriously when he shouts at me." (17-year-old married mother)*

Some participants reported that their partners tried to coerce them to abort the child because the unplanned pregnancy would bring shame to their families and that partners would not support them financially if they continued the pregnancy. Participants discussed

the belief among young men in rural areas that, if the girl is single and pregnant, the pregnancy belongs to more than one man with a few participants discussing that their partners felt this way and threatened not to provide support if they refuse to abort. However, these participants discussed being fearful of complications associated with having an abortion such as death and religious beliefs, as abortion is considered a grave sin. Some adolescent mothers reported that they tried to seek abortion services within the village as hospitals do not conduct such procedures because abortion is illegal in Malawi. "*Yes! I thought about aborting, but deep down was not up for it. The responsible man told me to abort, but I told him what if I die in the process of aborting... What will be my benefit from it? Maybe I should have problems during delivery, but I cannot abort.*" (15-year-old single mother)"*I had feared to abort, but the man insisted on aborting so that we continue our relationship. We had unprotected sex once, and he told me I could not get pregnant, but I got pregnant. So, I told him we should take care of the pregnancy.*" (16-year-old unmarried mother)

Unhappy marriages were mostly driven by forced marriages, where some adolescent mothers experienced powerlessness by being coerced into marriages, which resulted in psychological abuse, financial stress and pressure to conform. These influenced chronic stress and poor mental health.

## Single mothers or unmarried

A few adolescent mothers who were single reported they wished to get married. However, their partner refused. Their main concern was how they would be able to meet their child's needs. Some feared reducing fathers' involvement in the child's care. However, most of the single mothers in this study were unmarried and living with their relatives or parents and reported limited involvement of the father of the child. While a few participants experienced rejection by their families, many of the single mothers discussed how their parents accepted the situation and willingly supported them and their children. "*My parents accepted it. They started to give me all the support I need*" (15-year-old unmarried mother). Another adolescent mother said, "*I have gone back to school, and my parents look after my child. I have learnt my lesson.*"16-year-old unmarried mother)

The impact of social and cultural influences on relationships of adolescent mothers impacted positive mental wellbeing for some while for others caused distress and unhappiness. Cultural beliefs impacted early marriage and expectations for adolescents who became pregnant to marry. For some the stressors of being in a relationship exacerbated mental health issues, while for others this was a positive experience. For some remaining unmarried contributed to better wellbeing while for others, stressors around social expectations and financial pressures exacerbated mental health issues.

## Support from family

A few adolescent mothers referred to family dynamics impacting their psychological wellbeing. Married and single participants highlighted the importance of support from their families throughout pregnancy and delivery. While some participants expressed negative experiences such as feelings of rejection, isolation, and not being supported, others received positive support from their families.

## Limited family support

Some adolescent mothers discussed that, while they had assistance from family and friends, it was frequently inconsistent or insufficient. In their social context, they also mentioned receiving criticism and condemnation from parents and significant others, which made them feel

ashamed and alone, especially when they were unmarried. For some participants, this rejection resulted in minimal engagement with their parents which was a stressor. A few mothers discussed the rejection continuing after the birth of their baby, which resulted in diminished communication with family members and ongoing unresolved tension. For example, one adolescent stated, "*My parents were not happy. They told me to go to stay with my husband, and they took me to the man responsible for the pregnancy. My parents had nothing to do with me.*" *(16-year-old married mother) and* another adolescent mother said, "*They (parents) chased me out of their house, then my mother would shout at me because of that... So that worried me a lot*". (age and marital status of the mother?) "*My parents were not happy; I went to stay with my mother-in-law when I was 4 months pregnant... I have stayed here for about 7 months*" (.15-year -old mother)

Many adolescent mothers discussed feelings of self-blame and regret. A few discussed feelings of guilt associated with not listening to their parents and guardians and breaching family values and expectations. Therefore, their suffering was taken as a reward for their wrongdoing. Adolescent mothers expressed emotions of self-blame, believing that due to their mistakes, they lacked basic needs. This contributed to a missed opportunity for support. For example,

> I kept this to myself......I didn't feel like sharing this with anyone to avoid conflicts......I was just regretting why I did this to get into this problem......So, I did not share it with anyone, even my close relatives. *(17-year-old unmarried mother)*

> I feel embarrassed because my pregnancy was not planned, so if I need something, my relatives will say that this is what I wanted....So, I decided to keep it in my heart." *(18- year-old– mother)*

### Good family support

Most adolescent mothers voiced that social support from family members played a critical role in fostering their emotional well-being throughout the perinatal period; they appreciated the physical presence of their mother and emotional support through guidance and advice throughout the postpartum period. *"The parents waited until you are healed from the labor wounds and then they leave...that is the time you start doing some of the things by myself."* This support gave them a sense of security, belonging, and improved their self-esteem and wellbeing. The support provided by their parents was also viewed as an opportunity to learn parenting skills from their mothers. They also appreciated the financial support, especially those who were single.

Family support acted as a protective factor and contributed to enhanced wellbeing for adolescent mothers who received good support. These mothers benefited from the social and cultural influences whereby family members passed on parenting skills and were emotionally and financially supportive. However, rejection from the family negatively impacted participants' social and emotional well-being.

### Rite of passage

Rite of passage included three sub-themes: baby and delivery preparation ceremony, postnatal confinement, rituals, beliefs and superstitions. Within this theme, adolescent mothers described traditional ceremonies and customs that are followed before and after delivery to prepare for motherhood. Participation in these ceremonies and adopting cultural customs reduced anxieties related to childbirth.

## Baby and delivery preparation ceremony

Family members organizing ceremonies to prepare mothers for delivery and childcare was a commonly discussed event. Such ceremonies reduced mothers' anxiety about childbirth and caring for a baby. For example, a ceremony arranged by family members involved elderly women providing advice and counselling to adolescent mothers on how to prepare for labour, delivery and caring for the baby and their family. Most participants considered that this ceremony helped them prepare mentally for childbirth. Some participants reported that the ceremony allayed their anxieties and built confidence in them as new mothers. In addition, mothers reported that women also organized useful gifts for the baby, such as wraps, alleviating some concerns about material baby needs. For example:

> "*They organized women who advised me on marriage issues since we are now expected to behave like adults. . . of course, at different stages of the pregnancy, they would bring elderly women to advise me. They also came to advise us on how to deliver the baby; I was happy with this. I was not afraid when I had my child. I was given 16 wraps, yes, I have 16 of them and also people who came to see the baby brought some as gifts; I was ready emotionally* "(17-year-old married mother). "*In the village, they advise us about delivery and how to care for a child. So, after they advised me, the worries stopped. . .. They said you should not go against the doctor's advice but follow it. . .. I felt well prepared*". (18- year- old married mother)

In contrast, a few participants indicated that due to estrangement, their families did not organize any ceremonies. These participants discussed feelings of inadequacy and neglect. Culturally, it is usual for the mother's parents to arrange the ceremonies, and for some participants, this was not a role in-laws were financially prepared to take on, for example: "*My mother did not want. I told you my parents were not talking with me. I lived with my mother-in-law, who said that my parents should make the arrangement because she did not have money to pay the ladies; she did not have food to cook for them. I only heard what happens at the hospital from my friends. I know my in-laws did not want.*" (19-year-old married mother), Another adolescent who also faced a reflection said parents complained: "*They never told us anything about childcare; we only followed advice from the hospital. We just eloped and my parents were not staying close to us*".(16-year -old- married mother)

## Postnatal confinement

Most participants discussed the cultural expectation that after the birth of the child, they are encouraged to rest within their household with restrictions imposed on cooking, household chores, and attending social events for one week. After one week, mothers can go out to socialize, however, they are not allowed to cook for a month. During this period, they are accompanied by elders who sleep in the same room with the mother and the baby and assist with baby care and cooking. This postnatal practice is said to allow the body of the adolescent mother to recuperate from the physical trauma of the delivery process after childbirth. The practice was also linked to protecting the baby and their husband from other illnesses. Many participants expressed that, during this period, elders teach adolescent mothers how to take care of the infant while at the same time instilling confidence. The elders also aim to protect the mother and the baby from bad omens such as infant illness and death. The period of accompaniment varied from 6 weeks to 3 months. Failure to follow this tradition is believed to cause severe illnesses in the baby, the husband, and potentially death of the infant. During this period, mothers are expected to always be with the baby to protect them from witchcraft spells that may

target the baby's umbilical cord. So, they (elders or parents who live with the mother after delivery) try to guard the baby's cord until the cord comes off and follow all the protective rituals. Many adolescent' mothers acknowledged that this practice helps them to cope with child-caring as postnatal new mothers.

"*They (Elderly women) stay in my house they will leave today. . . They were cooking for me; they talked about family planning and contraceptive methods. They told us to stay for 3 days without anyone touching the baby except the mother during breastfeeding, and even up to now, no one should touch their baby. They help us to settle down and cope well until we are healed.*" *(17-year-old Mother)*

## Rituals, beliefs and superstitions

Adolescent mothers discussed several cultural rituals that were followed to protect the infant's life, with the most commonly reported "*kuika mwana kumalo*" "*or putting the baby in the right place, making the baby safe from bad things including sickness.*" Participants discussed feelings of anxiety and uncertainty regarding the baby's life if these rituals were not followed. Part of this ritual "kuika mwana kumalo "marks the time when couples have their first sexual contact after delivery (3–6 months post-delivery), with the aim of making their baby strong. This period of abstinence from sex is adhered to by the couple as well as the couple's parents. Not having a partner to perform this ritual with or having a partner who was not trustworthy during the abstinence period made mothers fearful of negative consequences. Participants without partners were also concerned about the life of their baby. Some adolescent mothers who did not follow these rituals feared losing their child due to illness. A few participants reported to have been forced to sleep with another man if they do not have a husband to fulfill the ritual after postnatal abstinence, subjecting them to sexual abuse and associated mental health problems.

"*We wait until the baby is about 4 months old and then you can start having sex with your spouse. . . Sometimes, you even wait for 5 months; we are supposed to strictly follow the rules on how to do it. Otherwise, the baby can be weak, not strong and may die.*" *(18-year-old married mother)*

"*Some say that we are supposed to be advised on what to do after childbirth on having sex to prevent the baby from getting sick, but my partner is not available. My in-laws told me to have sex with him. ha ha ha (laughing)*" *(16-year-old married mother)*

Some participants also described the belief that the adolescents' husbands or partners are expected to take some traditional medicine to protect the child from illnesses. This traditional practice was viewed as a cause of worry for some adolescents, especially in situations where the husband was unfaithful or in a polygamous relationship. These adolescents feared the loss of their baby and this fear affected their emotional health during the postnatal period.

The rituals, such as postnatal confinement, were often accompanied by increased social support and, therefore, reduced the burden of care while promoting the bonding of the infant with the mother, which was positive for good mental health. However, for other adolescent mothers, the superstitions and failure to adhere to rituals led to feelings of isolation, worry, guilt and fear which negatively impacted the adolescent's mental health

## Discussion

This study aimed to explore social and cultural factors that may influence the mental health of adolescent mothers. Three major themes emerged: 1) *health and mental health awareness*; ii)

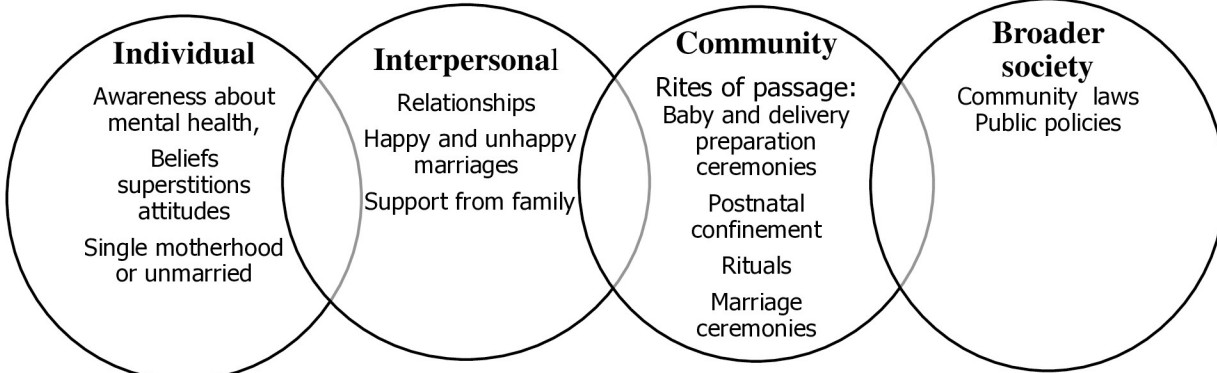

**Fig 2. The application of the SEM to understand social and cultural influences on adolescent mental health in Malawi.**

*relationships*, *support from family; and* iii) *rite of passage*. The Socio-Ecological Model (SEM) was used to understand the study findings with consideration of influences at individual, interpersonal, community levels and within broader society Fig 2 [49,50]. Individual level influences were *beliefs*, *attitudes* and *mental health awareness*. At the interpersonal level relationships, and family support influenced adolescent mental health. Community level influences encompassed social norms such as *rite of passage*, *pre-birth preparation ceremonies*, *postnatal confinement* and rituals and, in broader society, public policies impact individual behaviours. See Fig 2.

Within the overarching theme of *health and mental health awareness*, adolescent mother's understanding of health and mental health was found to be a characteristic that was influential in shaping adolescents' understanding and perceptions of mental health problems and the behaviors related to seeking help. Awareness is an individual-level factor [40,51]. It includes an understanding of maternal mental health issues, recognizing symptoms of mental health, and having knowledge surrounding good mental health that usually helps in shifting attitudes towards mental health [52]. A good understanding of mental health helps to reduce stigma, promote psychological wellbeing, and encourage help seeking [53]. In this study, many adolescents demonstrated limited understanding and awareness of possible contributing factors for good mental health, and some were reluctant to seek professional assistance. These findings are similar to a study conducted in Malawi among perinatal adults [54], and an Ethiopian study which found that 45% of community members do not recognize the symptoms of poor mental health and the need to seek help [55]. The Ethiopian study was a quantitative study (n = 420 which found that 88% of the participants felt mental health problems are attributed to possession by an evil spirit, 76.4% reported God's punishment, and 67.1% witchcraft, which impacted seeking support. Similar to our study, the study found low education levels, limited mental health literacy, and poor access to support may contribute to a lack of awareness and help-seeking [53]. Enhancing mental health literacy is likely to enhance awareness, help seeking and reduce community and individual stigma associated with mental health [53].

Interpersonal level [51,56] themes were *relationships and family support*. The two themes are interlinked within the interpersonal level, and they play a crucial role in influencing mental health of adolescent mothers. A subtheme of *relationships*, *positive and happy marriages* enhanced support for adolescent mothers throughout the perinatal period, having a positive influence on the adolescent's mental health. The support included financial, emotional, informational and companionship. Therefore, when mothers had disagreements with their families and husbands, they felt rejected. However, for some, the relationship between the adolescent

and their parents improved after childbirth, while others remained unsupportive which impacted their mental health. Similarly, studies conducted in Nepal and Mexico found that support from a caring family or partners acted as a protective factor in adolescent lives and could compensate for the absence of other protective factors, with the relationship centering on the care and needs of the child [57,58]. Furthermore, a Nepalese study among older perinatal women found that women are more likely to use maternal care services when their husbands accompany them [58,59]. The more the women considered their husbands were supporting them, the less likely they were to experience maternal anxiety [58], and the absence of a husband was an independent predictor for postpartum depressive symptoms [60]. This may also be true for Malawian mothers. It is clear from the findings of this study that many of the psychological problems inherent in adolescent motherhood were related to a lack of family or partner support, parental criticism, and possible abuse. These issues are similar to findings from a systematic review of psychosocial risk factors for perinatal depression among female adolescents and a Rwandan study among adolescent mothers n = 120 [61,62]. This study found that adolescents who were in polygamous relationships, forced marriages, lacking social support, experiencing intimate partner violence and those whose parents were not involved in decisions regarding their care were more likely to report common mental disorders [38,63]. For participants in the current study, societal norms mean many adolescent mothers depend on their parents and husbands for crucial decisions. Similarly, for some other cultural groups in SSA and Asia, decisions regarding reproductive health care are dependent on parents' or husbands' decisions [64,65]. Therefore, interventions should focus on enhancing positive relationships within the interpersonal spaces and create an environment that supports adolescents' choices and respect.

The theme *rite of passage* with subthemes *baby and delivery preparation ceremonies*, *postnatal confinement*, and *rituals beliefs and superstitions a*re associated with the 'community level 'or 'social cultural level' within the Social Ecological Model [40]. Cultural norms, traditions, and cultural expectations influenced mental health positively and negatively in this study. Social norms such as rite of passage ceremonies and conforming to rituals shaped adolescent mothers' compliance with their beliefs, values and attitudes. For example, the *pre-birth and delivery cultural ceremonies* described in this study provided support and reduced anxiety around birth and parenting. In this particular community, grandparents, parents, and aunties provide sexuality education. This is especially important as school-based sexuality education may be limited, especially given many adolescent mothers are not attending school [66]. While health workers may provide some education, this may also be limited [67]. Adolescent mothers in this study found these ceremonies enhanced their knowledge and skills which reduced concerns and stressors associated with birth and childcare. Similarly, a study in Ghana among adolescent mothers, where pre-birth ceremonies are also common, found these ceremonies assisted mothers in developing competencies to deal with postnatal challenges, and they felt prepared to take a new role as mothers [68]. In the current study, some of the rituals such as "kuika mwana kumalo" promoted rest and healing for mothers; however, some girls who were unmarried were subjected to sexual abuse, whereby adolescent mothers were forced by family to sleep with another man. Those who refused felt that they were unable to adhere to strongly prescribed cultural values, which resulted in distress and anxiety. For those who did comply, this type of ritual may subject adolescent mothers to further mental distress and place them at risk of HIV and other STIs [69] or another unplanned pregnancy [70]. Interventions that promote collaboration with community members and other social networks while respecting the cultural context can promote mental health and wellbeing. Codesigned interventions with adolescent mothers to address sexual abuse and to enhance assertiveness and negotiation skills in a culturally respectful manner would be beneficial.

At a societal level [71], findings from the study found some gaps in public policies to support adolescent mothers. For example, some participants discussed financial insecurities, which enhanced the likelihood of family conflicts and resulted in some adolescent mothers rushing into early marriage. In a similar setting, a study in Ghana among adolescent mothers found that access to social and economic support was associated with better outcomes for adolescents during motherhood including reduced risk of distress [68]. Public policies that address financial assistance, such as parenting payments as well as the provision of financial education for adolescent mothers, are imperative.

Adolescent mothers in this study also faced additional emotional and psychological challenges around abortion if the pregnancy was unplanned. The findings highlighted the challenges associated with legal restrictions on accessing abortion and safe reproductive health services combined with the social pressure that impacts adolescents' decisions about their unplanned pregnancies. Social and systemic challenges that adolescents experience in accessing reproductive services and the concerns around the safety of abortion services are public policy issues. These challenges are similar to a study that was conducted in Uganda among teen girls which found that access to comprehensive reproductive health services was limited [72]. To address these challenges, improving access to comprehensive reproductive health services through legal, funding and policy changes is important.

Considering Malawi continues to have the 11th highest rate of child marriage in the world, with half of all girls marrying before the age of eighteen [9], policy and public health relevance are crucial because of the documented risks associated with adolescent motherhood and the unmet needs for information and services. The complex nature of these practices and the lack of data documenting their impacts on adolescent mothers have historically been significant barriers to addressing these issues systematically. Therefore, this study has the potential to inform the development of appropriate mental health promotion strategies that are culturally appropriate. It will further inform policymakers of the need to improve policy guidelines on management of adolescent mothers.

## Strengths of the study

To our knowledge, this is the first study describing the understanding of social and cultural factors that influence the maternal mental health of adolescent mothers in rural areas of Malawi. The study presents adolescent mothers' own voices. Application of the Social Ecological Model emphasizes the importance of understanding the various interconnectedness of factors that are mutually influential on an individual's psychological wellbeing and helps plan where interventions can be focused. Furthermore, the use of qualitative design, with a large participant sample, enabled the collection of rich data from mothers with a lived experience [73].

## Limitations

The study was conducted during the COVID-19 pandemic, and this affected recruitment and data collection. Some participants were wary of recording the interviews due to conspiracy theories associated with COVID-19. For example, they believed that the researcher was satanic and wanted to impose COVID-19 vaccinations through an electronic chip that would track them through their voices. The study also involved adolescent mothers from one rural hospital catchment area, which may not represent all health facilities in the district or other rural areas of Malawi. Furthermore, the questions were part of a larger survey of diverse topics which meant the scope of discussion had some limitations. Marriage of girls aged under 18 is illegal in Malawi. Therefore, some of the mothers worried that their husbands might be arrested and were hesitant to discuss some issues in detail.

## Conclusion

The study highlights some of the social and cultural influences that influence the maternal mental health of adolescent mothers. The results provide a better understanding of possible opportunities for designing and implementing appropriate and targeted mental health promotion interventions for adolescent mothers and the public, considering all cultural sensitivities and taboos. By addressing factors at each level of the model, a comprehensive understanding of the complex and interconnected factors that may impact adolescent mothers' mental well-being can be promoted. Future research should examine the effectiveness of psychosocial interventions on adolescent mental health, and policies should focus on long-term interventions targeting the prevention of early marriages, improving access to reproductive health services, and including sexual health and mental health prevention in the school curriculum.

## Acknowledgments

The authors acknowledge the support of the head of the School of Population Health, Curtin University. the Lilongwe District Health Officer for his support, and the Clinic in charge for the coordination. We would also like to acknowledge the following: Precous Kachale (PK): Charles Ndawala (CN) for assisting with data collection, Marumbo Gondwe for transcribing (MG), and Maggie Zgambo for checking the codes (MZ). We are grateful to adolescent mothers who participated and gave their time and information.

## Author Contributions

**Conceptualization:** Chimwemwe Tembo, Linda Portsmouth, Sharyn Burns.

**Data curation:** Chimwemwe Tembo, Linda Portsmouth, Sharyn Burns.

**Formal analysis:** Chimwemwe Tembo, Linda Portsmouth, Sharyn Burns.

**Funding acquisition:** Chimwemwe Tembo, Sharyn Burns.

**Investigation:** Chimwemwe Tembo, Linda Portsmouth, Sharyn Burns.

**Methodology:** Chimwemwe Tembo, Linda Portsmouth, Sharyn Burns.

**Project administration:** Chimwemwe Tembo, Sharyn Burns.

**Resources:** Chimwemwe Tembo, Sharyn Burns.

**Software:** Chimwemwe Tembo, Sharyn Burns.

**Supervision:** Linda Portsmouth, Sharyn Burns.

**Validation:** Chimwemwe Tembo, Linda Portsmouth, Sharyn Burns.

**Visualization:** Chimwemwe Tembo, Linda Portsmouth, Sharyn Burns.

**Writing – original draft:** Chimwemwe Tembo.

**Writing – review & editing:** Chimwemwe Tembo, Linda Portsmouth, Sharyn Burns.

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
