## [Decision Letter · Decision Letter 0]

5 Nov 2024

PMEN-D-24-00096

“They organized women who advised me”: Social and cultural influences of adolescent mental health during the postnatal period in Malawi: Qualitative study

PLOS Mental Health

Dear Dr. Tembo,

Thank you for submitting your manuscript to PLOS Mental Health and we apologise for the severe delay in reaching a decision. After careful consideration and taking the reviewer comments into account, we feel that your study has merit but does not fully meet PLOS Mental Health’s publication criteria as it currently stands. Therefore, we invite you to submit a revised version of the manuscript that addresses the points raised during the review process.

Please ensure that you fully address all of the points raised by the reviewers. Given the fact that one of the reviewer comments is very brief, we may need to include another reviewer for the revision. However, as this manuscript is already very delayed, we wanted to give you the chance to work on the revisions that have already been suggested. Thank you for your patience with this process and I understand that it may have been frustrating. 

We look forward to receiving your revised manuscript.

Kind regards,

Karli Montague-Cardoso

Executive Editor

PLOS Mental Health

Journal Requirements:

 1. We note that you have indicated that there are restrictions to data sharing for this study. For studies involving human research participant data or other sensitive data, we encourage authors to share de-identified or anonymized data. However, when data cannot be publicly shared for ethical reasons, we allow authors to make their data sets available upon request. For information on unacceptable data access restrictions, please see http://journals.plos.org/plosone/s/data-availability#loc-unacceptable-data-access-restrictions.   Before we proceed with your manuscript, please address the following prompts:  a) If there are ethical or legal restrictions on sharing a de-identified data set, please explain them in detail (e.g., data contain potentially identifying or sensitive patient information, data are owned by a third-party organization, etc.) and who has imposed them (e.g., a Research Ethics Committee or Institutional Review Board, etc.). Please also provide contact information for a data access committee, ethics committee, or other institutional body to which data requests may be sent.  b) If there are no restrictions, please upload the minimal anonymized data set necessary to replicate your study findings to a stable, public repository and provide us with the relevant URLs, DOIs, or accession numbers. Please see http://www.bmj.com/content/340/bmj.c181.long for guidelines on how to de-identify and prepare clinical data for publication. For a list of recommended repositories, please see https://journals.plos.org/plosone/s/recommended-repositories. You also have the option of uploading the data as Supporting Information files, but we would recommend depositing data directly to a data repository if possible. Please update your Data Availability statement in the submission form accordingly. 2. Please provide an Author Summary. This should appear in your manuscript between the Abstract (if applicable) and the Introduction, and should be 150–200 words long. The aim should be to make your findings accessible to a wide audience that includes both scientists and non-scientists. Sample summaries can be found on our website under Submission Guidelines:  https://journals.plos.org/mentalhealth/s/submission-guidelines#loc-parts-of-a-submission 3. Please provide separate figure files in .tif or .eps format. For more information about figure files please see our guidelines:  https://journals.plos.org/mentalhealth/s/figures https://journals.plos.org/mentalhealth/s/figures#loc-file-requirements 

Additional Editor Comments (if provided):

Reviewers' comments:

Reviewer's Responses to Questions

**Comments to the Author**

1. Does this manuscript meet PLOS Mental Health’s publication criteria? Is the manuscript technically sound, and do the data support the conclusions? The manuscript must describe methodologically and ethically rigorous research with conclusions that are appropriately drawn based on the data presented.

Reviewer #1: Yes

Reviewer #2: Partly

2. Has the statistical analysis been performed appropriately and rigorously?

Reviewer #1: Yes

Reviewer #2: N/A

3. Have the authors made all data underlying the findings in their manuscript fully available (please refer to the Data Availability Statement at the start of the manuscript PDF file)?

Reviewer #1: Yes

Reviewer #2: No

4. Is the manuscript presented in an intelligible fashion and written in standard English?

Reviewer #1: Yes

Reviewer #2: Yes

5. Review Comments to the Author

Reviewer #1: A well-written manuscript. I was eager to know the outcome of the study. The description of each method was precise. It is an adaptable study approach that I would also consider applying in Zambia to assess if the outcomes would be similar or differ.

Reviewer #2: Comment

Title

• The title of the article is not reflective of the study objective and/or the core research question that the authors aim to address in this qualitative study.

Abstract

• The result section of the abstract needs more clarity on how the findings reflect the research objective i.e. to explore perceived cultural and social factors influencing mental health among adolescent women.

• It would be better if the author omits the Discussion section on the abstract and merged the findings portions into the result section and the conclusion to the conclusion section of the abstract

Introduction

• In the introduction section author has tried to illustrate the current status of adolescent pregnancy in the region followed by the legal provision and strategies adopted by the government of Malawi to control the issue of child marriage. Then the author moves to the potential consequences of adolescent motherhood and surgically connects it with the postnatal period. The author talks about the importance of seeking mental health support and early identification problems followed by setting the connection between cultural and social factors impacting mental health disorders.

o But nowhere in the introduction, the author has specified: what is the nature of the mental health problems that adolescent mothers are experiencing and what are the potential social and cultural factors that have been observed to affect maternal mental health in countries similar to Malawi. Though this is a qualitative study the author could look into some quantitative studies so they could reflect more on the current magnitude of maternal mental health conditions such as antenatal depression, postpartum depression, maternal blues, anxiety, stress, and others in the context of Malawi, Africa, and Asian context which share similar social and cultural attributes.

o There is a need to spell out the cultural and social factors that might affect maternal mental health so that international readers can better understand the context in which this study is executed.

o The introduction section states the aim of the study as “aims to explore how social and cultural factors influence the mental health of adolescent mothers in rural Malawi,” while the objective in the abstract section reads, “To explore the perceived cultural and social factors that influence mental health among adolescent mothers during the postnatal period in rural Malawi.” These two statements are quite similar but not entirely aligned in terms of focus. The introduction suggests a broader scope of exploring actual social and cultural factors, while the abstract narrows it down to perceived factors, specifically during the postnatal period. There is a need to maintain consistency in the abstract, introduction, and research question.

Methods

• It is suggested that the author could include the Theoretical framing as a part of the methods and not the introduction

• I would like to request the relevance of mentioning this statement in the method section “…….The facility is 128 enclosed by a brick fence, and just outside of the fence there is an open, crowded market with 129 many restaurants, some drug stores, pubs, and many herbalists selling different kinds of herbal 130 medicines.” If it is not connected with the study in any way then I would like to recommend the author remove irrelevant contents throughout the manuscript to keep it concise and specifically tailored to the stated research question.

• In the method section, under Study Design the author has provided valuable information about the researcher’s background and its potential impact on the study. However, to enhance clarity and focus, consider streamlining this portion. For instance, while the researcher’s cultural background is crucial, details about their age and specific personal experiences might be less relevant. A more concise explanation emphasizing the researcher’s role and steps taken to ensure objectivity could be beneficial. This would help maintain the reader’s focus on the study’s design and methodology.

• In the participants recruitment and sampling, it is mentioned that “….. participants were adolescent mothers aged 19 and younger from the catchment area of Mitundu Hospital”, it is unclear if this means those living near Mitundu Hospital or those who came for an OPD visit in Mitundu Hospital or the mothers inside inpatient department were also taken. In the next line, it is stated that “Recruitment occurred when mothers attended a clinic for postnatal check-ups, under-five clinics, and family planning clinics at Mitundu Rural Hospital.” It is recommended that the author remove the redundancy and unwanted explanation to avoid confusion

• “..Interested adolescents meeting the eligibility criteria were provided with an information sheet” What were the eligibility criteria?

• “Participants completed a survey and in-depth interview detailed in previous publications (Tembo et al., 2022; Tembo et al., 2023)” This statement suggests that this is a secondary paper. But in the introduction section or methodology, there is no mention of this case. For better transparency, it would be better if the author stated the nature of the primary study, how this research question was derived from this original study, and how it differs.

• Were all 395 interviews transcribed by MG? How many days does it take for the author to transcribe the interviews?

• The authors mentioned that NVivo version 11 was used. Were all of the coding and analysis done manually or with the use of NViVo?

• Were the transcripts sent to the participants for their reflection?

Result

• Suggest to change the heading Findings to Results.

• Is there any specific reason to mention this in the result section “Three hundred and ninety- five interviews were conducted with adolescent mothers.” This is already stated in the method section

• With mean it is statistically sound to express the SD as well. What was the minimum and maximum age range of the adolescent mothers?

• Table 2 ‘N’ should be noted as ‘n’

• In the overall result section, it felt like the stated themes, codes, and verbatim are not tailored to the specific research question i.e. what cultural and social factors influence mental health among adolescent mothers in rural Malawi? The result needed to be revised to make it concise and tailored to the stated research question.

• The presentation of themes in the results section could be clearer. Specifically, the "overarching theme awareness" and how it influences other themes, such as "relationships," "family support," and "rites of passage," needs more explanation. It would be helpful to explicitly describe how each of these themes interrelates with maternal mental health.

• The cultural context of the findings is not fully explored. The influence of specific cultural practices, beliefs, or norms related to motherhood, mental health, or adolescence in rural Malawi should be expanded upon. The connection between these practices and adolescent mothers’ mental health is essential and could strengthen the overall findings.

• While the quotes are compelling, their connection to the broader themes should be made clearer. Currently, they seem somewhat isolated from the discussion of themes. Each quote should be explicitly tied back to the theme or subtheme it supports. This would make the narrative flow more coherent and strengthen the findings.

• It would be better to reflect some anonymous socio-demographic information of participants in the quotes, so the readers could get to explore the difference in the expression across age, married/unmarried status, level of literacy, and so on.

• The results section lacks an in-depth exploration of how specific social and cultural factors (e.g., rites of passage) affect mental health. The connection between these factors and adolescent motherhood is vital but not sufficiently detailed.

Discussion

This discussion provides a comprehensive exploration of the social and cultural factors that influence adolescent maternal mental health in rural Malawi, framed within the Socio-Ecological Model. But certain improvements can be made:

• In the discussion, it is stated that “Within the overarching theme health and mental health awareness, individual ‘s understanding of health and mental health was found to be a characteristic that was influential in shaping adolescents’ understanding and perceptions of mental health problems and the behaviors related to seeking help.” However in the result section it is not established that the awareness is connected with or is influencing the perception of mental health or the occurrence of the mental health outcome in the adolescent.

• While the author references some studies from Nepal, Mexico, and Ethiopia, it would be useful to draw more detailed comparisons between the findings of the current study and other studies on adolescent mental health.

• The author should consider including quantitative studies from these countries in the Introduction section to provide a clearer picture of the magnitude of maternal mental health issues. Additionally, in the Discussion section, the author could enhance their analysis by comparing their qualitative findings with cultural and social factors related to maternal mental health identified in past qualitative as well as quantitative research. This will help contextualize the current study's findings within the broader body of research.

In terms of Ethical considerations, it is unclear how confidentiality and privacy were maintained. As for the minor's consent, it is mentioned that legal guardians witnessed the adolescent mother’s consent, were the interviews with the adolescent mothers taken in the presence of their mothers and/or legal guardians? It is mentioned that “Any woman who reported suicidal ideation was counseled and offered a referral for clinical mental health assessment and treatment” but throughout the result section the information regarding suicidal ideation seems to be missing.

6. PLOS authors have the option to publish the peer review history of their article (what does this mean?). If published, this will include your full peer review and any attached files.

**Do you want your identity to be public for this peer review?** For information about this choice, including consent withdrawal, please see our Privacy Policy.

Reviewer #1: **Yes: **Mah Wasi Asombang

Reviewer #2: No

---

## [Decision Letter · Decision Letter 1]

26 Dec 2024

“They organized women who advised me”: Social and cultural influences of adolescent mental health during the postnatal period in Malawi: Qualitative study

PMEN-D-24-00096R1

Dear Mrs Tembo,

We are pleased to inform you that your manuscript '“They organized women who advised me”: Social and cultural influences of adolescent mental health during the postnatal period in Malawi: Qualitative study' has been provisionally accepted for publication in PLOS Mental Health.

Best regards,

Karli Montague-Cardoso

Staff Editor

PLOS Mental Health

Reviewer Comments (if any, and for reference):

Reviewer's Responses to Questions

**Comments to the Author**

1. If the authors have adequately addressed your comments raised in a previous round of review and you feel that this manuscript is now acceptable for publication, you may indicate that here to bypass the “Comments to the Author” section, enter your conflict of interest statement in the “Confidential to Editor” section, and submit your "Accept" recommendation.

Reviewer #2: All comments have been addressed

2. Does this manuscript meet PLOS Mental Health’s publication criteria? Is the manuscript technically sound, and do the data support the conclusions? The manuscript must describe methodologically and ethically rigorous research with conclusions that are appropriately drawn based on the data presented.

Reviewer #2: Partly

3. Has the statistical analysis been performed appropriately and rigorously?

Reviewer #2: I don't know

4. Have the authors made all data underlying the findings in their manuscript fully available (please refer to the Data Availability Statement at the start of the manuscript PDF file)?

Reviewer #2: Yes

5. Is the manuscript presented in an intelligible fashion and written in standard English?

Reviewer #2: Yes

6. Review Comments to the Author

Reviewer #2: Thank you for the revision made which has improved the manuscript significantly

7. PLOS authors have the option to publish the peer review history of their article (what does this mean?). If published, this will include your full peer review and any attached files.

**Do you want your identity to be public for this peer review?** For information about this choice, including consent withdrawal, please see our Privacy Policy.

Reviewer #2: No
